# Testing the Greenhouse Emission Model (GEM) for Pesticides Applied via Drip Irrigation to Stone Wool Mats Growing Sweet Pepper in a Recirculation System

E. Louise Wipfler [1],*, Jos J. T. I. Boesten [1], Erik A. van Os [2] and Wim H. J. Beltman [1]

1 Wageningen Environmental Research, P.O. Box 47, 6700 AA Wageningen, The Netherlands
2 Wageningen UR, Greenhouse Horticulture, P.O. Box 644, 6700 AP Wageningen, The Netherlands
* Correspondence: louise.wipfler@wur.nl; Tel.: +31-3174-82875

**Abstract:** Pesticide emissions to surface water from greenhouses with crops grown on substrates in open or closed systems may be significant. It is important, therefore, to test models such as the Greenhouse Emission Model (GEM), which was developed to assess these emissions as part of the Dutch authorization procedure for use of plant protection products in greenhouses. GEM was tested using an experiment in which imidacloprid and pymetrozine were applied via drip irrigation to stone wool mats growing sweet pepper. The irrigation system in such greenhouses consists of a mixing tank to prepare the nutrient solution and a series of tanks to treat and recirculate the drain water back to the mixing tank. Emissions may occur because (part of) this recirculation water may be discharged or leached to the surface water. GEM assumes that all tanks are perfectly mixed. GEM further assumes that the water in these mats is perfectly mixed and that the pesticide behavior can be simulated by assuming one perfectly mixed reservoir. The model predicted breakthrough of both pesticides out of the mats earlier than measured, and the measured maximum concentrations were approximately two times lower than predicted. We considered a series of possible causes, including a smaller water volume in the mats, a higher plant uptake factor, and sorption to the stone wool. The model performance improved by representing the mats as a sequence of two equally large tanks with plant uptake restricted to the first tank. We recommend to study the solute transport process and the distribution of plant roots in the mats in more detail to further underpin the hypothesis used and improve the model. After this first validation, the GEM model might also be used in other countries to forecast emissions of PPPs to surface water.

**Keywords:** pesticide emission to surface water; greenhouse emission model; model testing; drip irrigation; soilless cultivation; pymetrozine; imidacloprid

## 1. Introduction

In the Netherlands, the area of greenhouses grown with vegetables or flowers is currently approximately 10,000 ha, of which approximately 8500 ha are soilless growing systems with substrates such as stone wool, peat, perlite, and coir [1,2]. The surface area of greenhouses is only a small fraction of the total Dutch agricultural area (nearly 2 million ha [3]). However, pesticide monitoring data for Dutch surface water have shown that pesticide use in greenhouses led to more exceedances of the acceptable concentrations in surface water than any of the agricultural land uses [4]. This may be partly caused by the higher pesticide use in terms of kg per ha for crops grown in greenhouses than for field crops [5]. Another cause may be that for the same application rate (dose), pesticide emissions from greenhouses to surface water are higher than from agricultural fields. Although excess irrigation water from soilless growing systems is reused (i.e., recirculated), part of this recirculation water may be emitted to the surface water to warrant good quality of irrigation water and prevent, for example, the sodium concentration from becoming too high [6]. Therefore, the assessment of pesticide emissions from soilless growing systems

in greenhouses is an important aspect of the Dutch pesticide registration procedure. The Greenhouse Emission Model (GEM) was developed to assess such emissions and calculate pesticide concentrations in the water that is discharged to nearby surface water [7]. The EU Water Framework Directive (EU, 2000) aimed to have ecologically and chemically sound surface water in 2015, with an additional 12 years if goals could not be achieved. The year 2027 is the target year for Dutch growers to have achieved the sound water quality goal as indicated in the WFD, and GEM is a tool to support the achievement of that goal. Although rural situations in other parts of Europe are often different, there is great interest in applying GEM in other regions with a high density of greenhouses or large areas of surface water.

For model acceptance in the pesticide registration procedure, it is important that the model has been tested against experimental data. The GEM model consists of a sub-model for the greenhouse water flows (the Waterstreams model, earlier described and tested [8]) and a sub-model for simulating the pesticide behavior in the greenhouse (called SEM: Substance Emission Model). The concentration in the water course to which excess water is discharged is calculated thereafter with a surface water model that simulates the solute transport, degradation, and adsorption processes in the water course. This work aims to test the SEM sub-model. An earlier study showed that an adequate test of SEM requires that the water flows in the experiment are measured in detail [9]. This work provides a test of the SEM model based on an experiment in which water flows were measured frequently and at various locations within the greenhouse [10].

There is a variety of growing systems (vegetables, pot plants, and flowers) and substrate types (stone wool, perlite, coir, and lava) used in greenhouses. The combination vegetables–stone wool, which covers approximately 4000 from the 8500 ha soilless growing systems in the Netherlands[2], was selected for the test. Here, a full nutrient solution is given to the plants, with the surplus (20–30%) being collected and reused after disinfection. Due to unbalanced nutrient compositions and accumulation of non-absorbed ions (sodium and chloride), part of the solution (1–10% annually) was discharged to the surface water [6]. Pesticides can be applied via spray application, low volume misting, or drip irrigation. We selected the application via drip irrigation. Drip irrigation leads to high emission concentrations because the complete dose is applied to the recirculating water, whereas for spray or low volume mister applications (LVM), only a fraction of the dose ends up in the recirculating water and, consequently, in the surface water. Two pesticides were studied, which are applied in commercial practice via the irrigation water, i.e., imidacloprid and pymetrozine. In 2016, these substances were among two of the most frequently used substances in greenhouses applied via drip irrigation. Monitoring data on surface water concentrations show that both substances are found in water courses near greenhouses and that imidacloprid is found at concentrations above the water quality threshold [4]. Note that both substances are no longer approved for use as plant protection products in open and closed cultivations in the Netherlands.

## 2. Materials and Methods

### 2.1. Experimental Procedures

The experiment was executed in a greenhouse of Wageningen University and Research in Bleiswijk (the Netherlands). The greenhouse was climate controlled based on incoming radiation and other weather parameters. The experimental compartment had a surface area of 144 m$^2$ and contained 12 rows of plants. In each row, 25 plants were grown at a distance of 40 cm in stone wool mats (Grodan Grotop Expert; $100 \times 12 \times 7.5$ cm) which were surrounded by plastic foil. This was realized by growing three plants on each mat at a distance of 10, 50, and 90 cm from the start of a mat and with a distance of 20 cm between the mats (Figure 1).

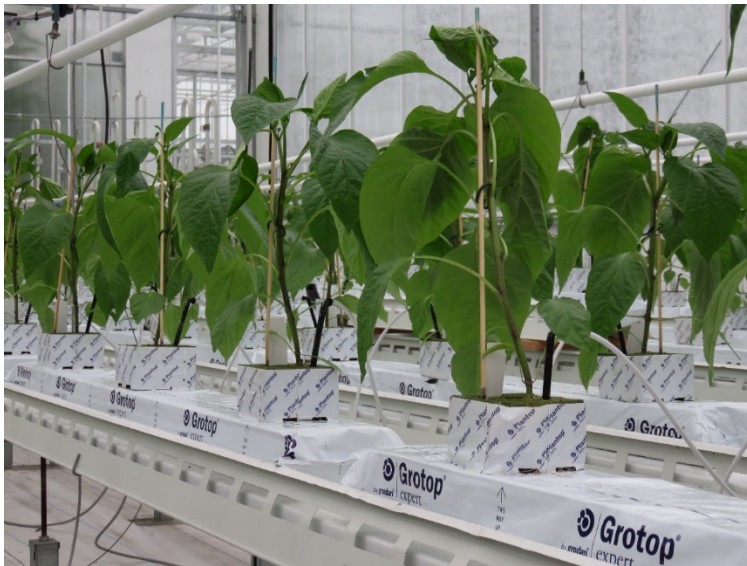

**Figure 1.** Blocks containing sweet pepper plants placed on top of the stone wool mats in holes in the plastic foil; photo taken before the start of the experiment. Each plant receives irrigation water, pesticide, and nutrients via drip irrigation. Water is collected in drain troughs below the stone wool mats.

Sweet pepper plants (cultivar Marinello) were raised at another location in stone wool blocks with sides also surrounded by plastic foil (Grodan Plantop Delta; $10 \times 10 \times 7.5$ cm). On 7 January 2016, blocks containing seven-week-old plants were transferred to the experimental compartment and placed on top of the stone wool mats in holes in the plastic foil (Figure 1). The experiment began (and pesticides were applied) on 31 May 2016, when the sweet pepper was full grown. The last harvesting date of sweet pepper was early November 2016, so only one growth cycle was considered.

The plants received water via drip irrigation (each plant had one pressure compensated dripper of 3 L/h). The irrigation water was prepared in a mixing reservoir (Figure 2). This included addition of nutrients and adjustment of the pH to 6.2. The water was added to the stone wool mats through PVC pipes (driplines) with an inner diameter of 40 mm and PE (poly-ethylene) pipes with an inner diameter of 16 mm. These drippers were connected to the 16 mm PE pipes by flexible PE tubes with an inner diameter of approximately 4 mm. The day before application of the two substances, i.e., on 30 May 2016, the irrigation was stopped at 16.00 h to obtain a relatively low water content of the mats at the time of application. This is common practice in Dutch greenhouses to obtain a more efficient uptake of the applied pesticide. On the first day, all irrigations were carried out manually, facilitating easy sampling. The volume of the first irrigation was 76 L (5 min) for the entire greenhouse compartment; it took place immediately after application, and it was followed by irrigation volumes of approximately 30 L (2 min) each hour. On the second day, the irrigation scheduling was automatically driven by global radiation; each dripper supplied 100 mL per 200 J/cm$^2$ radiation.

The base of irrigation water was a mixture of rain water that was collected from the roof of the greenhouse, reverse osmosis water, and drain water from the stone wool mats. Drain water from the mats was collected via coated metal troughs and flowed through PVC pipes to a sequence of reservoirs, as shown in Figure 2. The first reservoir was the filtration unit in which the drain water was filtered through a 3 μm fiber filter. The water was pumped from this unit to the so-called used-water reservoir in batches of approx. 35 L. The water was pumped out of the used-water reservoir in batches of approx. 38 L, i.e., the treatment volume of the subsequent ozone treatment unit. The ozone unit from Agrozone works as a batch reactor in which the water is treated with a redox value of 800 mV. As a consequence, the water level in the used-water reservoir changed in discrete

steps. It appeared that the ozone treatment degraded both pymetrozine and imidacloprid completely; the measured concentrations in treated water were always below the detection limit. After the treatment, the water was collected in a reservoir for cleaned water and then made available for reuse (application of nutrients up to a certain electrical conductivity).

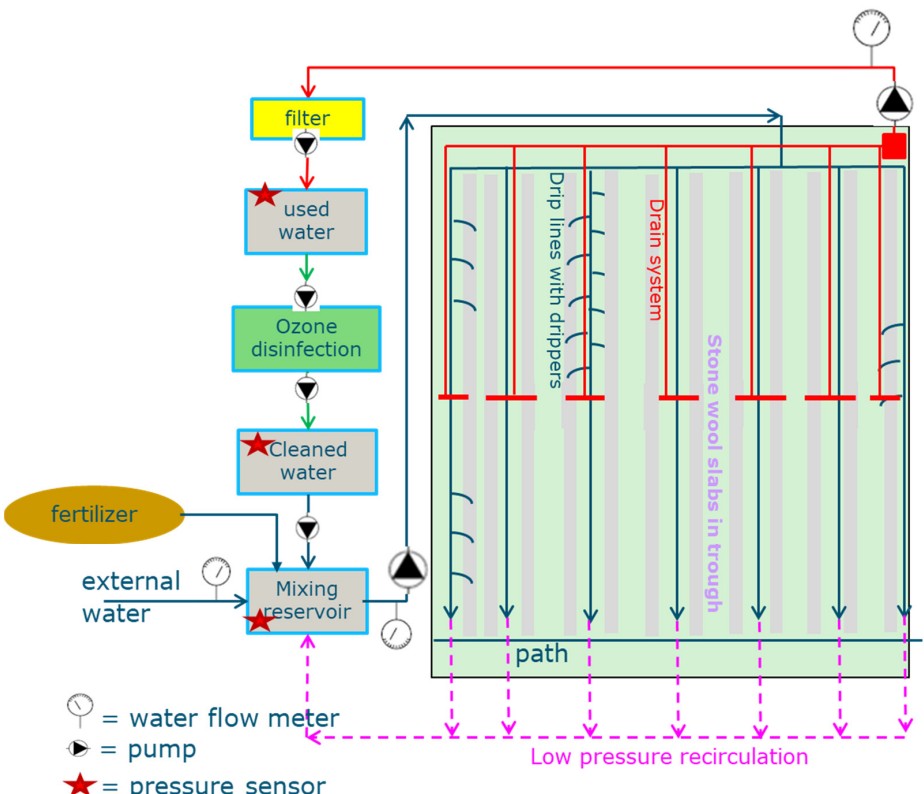

**Figure 2.** Schematization of the water flows in experimental compartment (surface area 144 m$^2$). External water (rainwater and osmosis water) is added to the mixing reservoir, in which it is mixed with nutrient solution by a fertilizer system. From the mixing reservoir, it is pumped to the dripping system via drip lines to the plants. The drain water is collected via troughs in the drain system and added to the recirculation system. Before reuse, the water is filtrated and disinfected. The pesticides are added to the mixing reservoir. The low-pressure recirculation is installed to make sure that all plants receive water with the same mixture of nutrients and with the same concentration of pesticides.

Floaters determined whether renewed filling of the mixing reservoir was required. After each renewed filling of the mixing reservoir, the solution was circulated under low pressure (below 0.8 bar) through the pipe lines, flowing back to the mixing reservoir in order to achieve a constant mixture of nutrient solution over the dripline system. During this low-pressure circulation (1–5 min), the so-called pressure-compensated drippers were closed. The advantage of using a drip irrigation system with low-pressure circulation is that all plants receive water with the same mixture of nutrients and with the same concentration of pesticides.

As shown in Figure 2, water volumes were measured in the mixing reservoir, the used-water reservoir, and the cleaned-water reservoir every 5 min with automatic pressure sensors. The volume of water in the cultivation compartment was based on measurements of the water content of the stone wool, which was measured every 3 min in duplicate with Grodan frequency–domain water sensors (based on measurement of the dielectric constant in the stone wool. The volumetric water content is highly correlated with the dielectric constant). These sensors consisted of three metal pins with lengths of 6 cm, which were horizontally placed in the stone wool in the middle between two plants. It resulted in 5–7 irrigations per day and a drain percentage of the surplus of 30%. The cumulative

water flow between the mixing reservoir and the cultivation compartment and between the cultivation compartment and the filtration unit were measured every 5 min with water volume counters. In addition, the volume of added rainwater was measured. During the experiment, there was no discharge of recirculation water to the surface water. Air temperature in the greenhouse was measured every 5 min.

The pesticides were applied to the mixing reservoir at approximately 10 h on 31 May as water-dispersible granulates, which contained 204 L of water. This included 41 L of water in the pipes used for the circulation of water under low pressure. Granulates were dissolved in 1 L water before being added to the mixing tank. A mass of 2.25 g of pymetrozine was applied as the formulated product Plenum (containing 50% pymetrozine), and a mass of 2.94 g imidacloprid was applied as the formulated product Admire (containing 70% imidacloprid). Imidacloprid is a neonicotinoid with the molecular formula $C_9H_{10}C_1N_5O_2$. Pymetrozine is a neuroactive insecticide and a member of the class of 1,2,4-triazines. It has the molecular formula $C_{10}H_{11}N_5O$. These application amounts were in line with the recommended doses on the label. Initial concentrations in the mixing reservoir, which were measured in duplicate, were 13,204 μg/L and 16,002 μg/L for imidacloprid and 8579 μg/L and 10,741 μg/L for pymetrozine. Samples for analysis of pesticide concentrations were taken in duplicate from the mixing reservoir and the used-water reservoir every two hours during working hours on the first two days. In addition, samples from the clean water reservoir were taken. Concentrations in these samples were all below the detection limit. The first sampling of the mixing reservoir took place immediately after application, i.e., before the circulation under low pressure (see above) took place. After two days, samples were taken only once a day. Samples were transferred to the lab and stored both prior and after analysis in a refrigerator at 4 °C (range: 2 to 8 °C). All samples were analyzed by reversed-phase liquid chromatography–tandem mass spectrometry (LC-MS/MS) after dilution with methanol: ultrapure water (15/85, *v/v*). The analyses were performed on an Agilent 1260 Infinity liquid chromatograph coupled with a 6460 Triple quad mass spectrometer (LC-MSMS) and equipped with Agilent jet stream electrospray ionization source (AJS-ESI) (Agilent Technologies, Santa Clara, California, USA).

Separations were carried out on an Agilent Eclipse XDB C18 column (4.6 × 150 mm, 5 μm) at 40°. The injection volume of the samples was set to 40 μL. The mobile phase used was Milli-Q water with 0.1% Formic acid (C) and MeOH with 0.1% Formic acid (D), with the following gradient: 0–5 min: 70/30 (C/D, *v:v*); 5.00–5.20 min: from 70/30 (C/D, *v:v*) to 10/90 (C/D, *v:v*); 5.20–8.20 min: hold on 10/90 (C/D, *v:v*); 8.20–8.30 min: from 10/90 (C/D, *v:v*) to 70/30 (C/D, *v:v*); and 8.30–11.00 min: hold on 70/30 (C/D, *v:v*) at flow rate of 0.5 mL/min. The mass spectrometer was operated using AJS-ESI in the positive mode. Nitrogen was used both as nebulizer and collision gas, the capillary voltage was 3500 V, and the temperature of the ion source was set to 300 °C.

The compounds were detected in the multiple reaction monitoring (MRM) using two transition per compound: imidacloprid 256.1/209 *m/z* and 256.1/175.1 *m/z* and pymetrozine 218.1/105 *m/z* and 218.1/78.1 *m/z*. Retention time was 3.6 min. for pymetrozine and 9.5 min. for imidacloprid.

Injected samples were quantified by peak area using a linear and forced-through-the-origin (*x*-axis zero; *y*-axis zero) calibration curve constructed from external standards included in the same sample sequence. Agilent Masshunter software was used for instrument control and data acquisition.

The detection limits (LOD) of imidacloprid and pymetrozine were 0.04 μg/L and 0.03 μg/L, respectively. The limit of quantifications (LOQ) were 0.12 μg/L and 0.10 μg/L, respectively. All data were collected in a free accessible experimental dataset [11].

### 2.2. Model Description

The SEM sub-model conceives of a greenhouse as a number of interconnected reservoirs that exchange water and solutes and in which each reservoir the water is perfectly mixed. This seems a priori defensible for the mixing reservoir, the filtration unit, and the

used-water reservoir, as these are water tanks. However, this can be called into question for the cultivation compartment (i.e., the stone wool mats), as the water flow through the mats is driven by gravity and by suction from the plant roots, which is likely to result in a solute movement process that differs from complete mixing. Nevertheless, complete mixing was assumed (as a starting point), this being the simplest approach possible and because no further information on flow processes in drip-irrigated rooted stone wool mats was available. Plant uptake was assumed to be proportional to the transpiration rate of the plants and the pesticide concentration in the water using the concept of the so-called transformation stream concentration factor (TSCF) [12]. TSCF indicates the efficiency of the translocation of a chemical in a root. The conservation equation for the mass of pesticide in each tank with number $i$ with upstream tanks $j$ and downstream tanks $k$ is then given by

$$\frac{d\,m_i}{dt} = +\sum_{j=1}^{\nu} Q_{fl,j,i}\,c_{w,j} - \sum_{k=1}^{\lambda} Q_{fl,i,k}\,c_{w,i} - V_{w,i}\,k_{t,i}\,c_{w,i} - Q_{up,i}\,TSCF\,c_{w,i} \quad (1)$$

where $m_i$ is the mass of pesticide in tank $i$ (kg), $\nu$ is the number of incoming water fluxes, $Q_{fl,j,i}$ is the volume rate of water flow (m$^3$/d) from tank $j$ to tank $i$, $Q_{fl,i,k}$ is the volume rate of water flow (m$^3$/d) from tank $i$ to tank $k$, $c_{w,j}$ is the mass concentration of pesticide in the water of tank $j$ (kg/ m$^3$), $\lambda$ is the number of outgoing water fluxes, $c_{w,i}$ is the mass concentration of pesticide in the water of tank $i$ (kg/m$^3$), $V_{w,i}$ is the volume of water in tank $i$ (m$^3$), $k_{t,i}$ is the rate coefficient of transformation of the pesticide in tank $i$ (d$^{-1}$) assuming first-order kinetics, where $k_{t,i} = \mathrm{Ln}(2)/\mathrm{DT50}$ and DT50 (d) is the transformation half-life of the pesticide, $Q_{up,i}$ is the volume rate of uptake of water by plant roots (m$^3$/d) which is zero for all tanks except the cultivation tank, and *TSCF* is the transpiration stream concentration factor of the pesticide (-).

Figure 3 shows the model configuration as it was tested against the experimental data. Each reservoir had only a single outgoing flux to another tank so no summation of the outgoing mass fluxes in Equation (1) was needed in this model test.

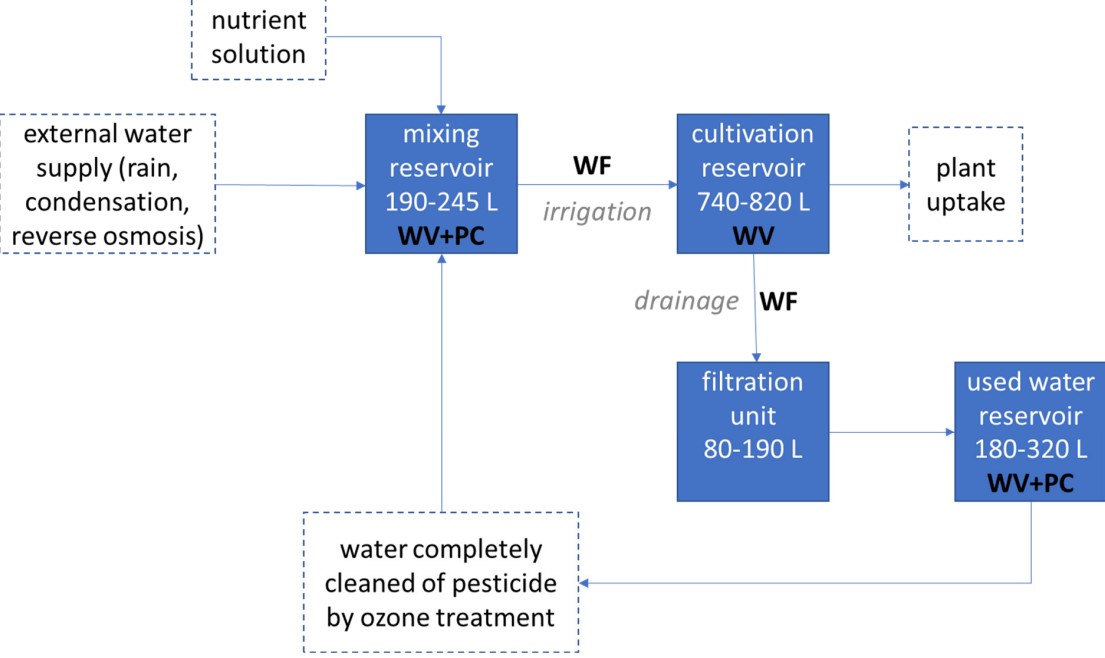

**Figure 3.** Schematic representation of the greenhouse system. The numbers indicate the range of the volumes of water in the reservoirs, WV indicates that the water volume in the reservoir was measured, WF indicates that the water flow rate was measured between the reservoirs in total volume per 5 min, and PC indicates that the pesticide concentration was measured in the reservoir. The blue boxes are considered in the model testing.

The mass of pesticide in each tank $m_i$ equaled $V_{w,i} * c_{w,i}$, so adsorption to the stone wool or any other material was not considered. The rate coefficient of transformation $k_{t,i}$ was assumed to increase with temperature following the Arrhenius equation (see [7]).

### 2.3. Model Parameterization

As described before, water flows and volumes were measured. However, some processing and interpretation of the measurements was needed to transfer them into a complete set of water volumes and water flow rates required for the model test.

The time course of the water volume in the cultivation reservoir was derived from duplicate measurements of the water content in the stone wool mats using the average of these measurements. As described before, the stone wool growing system consisted of mats (height 7.5 cm and volume 9 L each) on top of which three blocks were placed (height 7.5 cm and volume of 0.75 L each, so 2.25 L in total). Measurements of pF curves of stone wool [13] show that stone wool loses most of its water when the suction pressure of the water increases from zero (i.e., saturated) to 20 hPa: the volume fraction of water at saturation is approximately 0.98, whereas it is only approximately 0.20 at a suction pressure of 20 hPa. During most of the time, the bottoms of the stone wool mats were saturated (i.e., at zero suction, leading to drainage flow), whereas the top of the blocks (15 cm higher than this bottom) may have had a suction pressure close to 15 hPa (1 hPa corresponds to a pressure of a water layer of 1 cm). It is likely, therefore, that the volume fraction of water in the blocks is considerably lower than that in the mats. Thus, it was assumed (as a best guess) that the volume fraction of the water in the blocks was half that of the mats. So, the measured volume of water was based on a combined mat plus blocks volume of 10.12 L instead of the total rock wool volume of 11.25 L. This estimation procedure indicates that the estimated volume of water in the cultivation reservoir is somewhat uncertain (the possible effect of this uncertainty on pesticide behavior will be addressed later).

As indicated in Figure 2, the inflow of water into the mixing reservoir was not measured. However, it could be derived from the time courses of the water volume in and the water outflow from this reservoir. The water uptake rate of the crop was derived from the water balance of the cultivation reservoir (by combining the difference between inflow and outflow rates with the change in water volume in the mats). In addition, the water volume and the water outflow of the filtration unit were not measured. This water outflow could be derived from the stepwise increases in the water volume of the used-water reservoir. The water volume of the filtration unit could be derived from the difference between the measured inflow and the estimated water outflow (after deriving the initial volume via a measurement of the water height). The outflow of the used-water reservoir could be derived from the stepwise decreases in the water volume in this reservoir. Thus, a complete set of time courses of flow rates and water volumes (changing every 5 min) could be derived. The test of the model was limited to these four days.

The temperature in the cultivation reservoir was assumed to be equal to the air temperature in the greenhouse.

The dosages of the pesticides were based on the masses added to the mixing tank as described in the experimental procedures. The TSCF depends on the lipophilicity of a compound as shown by Briggs et al. (1982) [12]. It was estimated from the octanol–water partition coefficient using the equation:

$$TSCF = 0.784 e^{-[(\log K_{ow} - 1.78)^2 / 2.44]} \tag{2}$$

This returned 0.43 for imidacloprid on the basis of its octanol water coefficient ($K_{ow}$) of 3.7 [14] and 0.16 for pymetrozine on the basis of its $K_{ow}$ of 0.646 [15]. The half-life for transformation in the water was based on available hydrolysis half-lives. The hydrolysis half-life of imidacloprid was set at 1000 d at 25 °C because imidacloprid is reported to be stable [12]. The hydrolysis half-life of pymetrozine is 5–12 d at 25 °C and pH = 5, and it is reported to be stable at pH = 7. The half-life was assumed to be the average of 5 and 12 d, so

8.5 d at 25 °C. This may overestimate the hydrolysis transformation rate somewhat because the pH of the mixing tank was kept at 6.2. The molar enthalpy of the transformation rate (input to the Arrhenius equation) was assumed to be 65 kJ/mol (based on that for transformation in soil in the absence of better information [16]).

## 3. Results

The air temperature in the greenhouse showed a diurnal pattern with daily minima of 18–20 °C and daily maxima of 23–28 °C; daily average temperatures were 21–23 °C. Figure 4 shows that irrigation was restricted to the daytime (driven by the requirement of 100 mL irrigation for each dripper per 200 J/cm$^2$ radiation, as described before). The average daily irrigation volume was approximately 300–500 L, which corresponds to a water layer of approximately 2–4 mm for the 140 m$^2$ surface area of the compartment. The figure shows also that the drainage amount was, on average, approximately 30% of the irrigation amount, which is according to grower practices. The time of the start of the drainage outflow was closely linked to the time of irrigation inflow: detailed inspection showed that drainage started typically at approximately 1 h after the start of irrigation (please note that this does not mean that the residence time of a droplet of irrigation water in the cultivation unit is approximately 1 h: the irrigation induces a downward water flow which likely leads to drainage of water that was already present at the bottom of the mat). Detailed inspection revealed also that the first drainage occurred approximately 2 h after the first irrigation event and approximately 1 h after the second event. This 1-h delay after the first event was the result of the relatively low water content of the mats at the start.

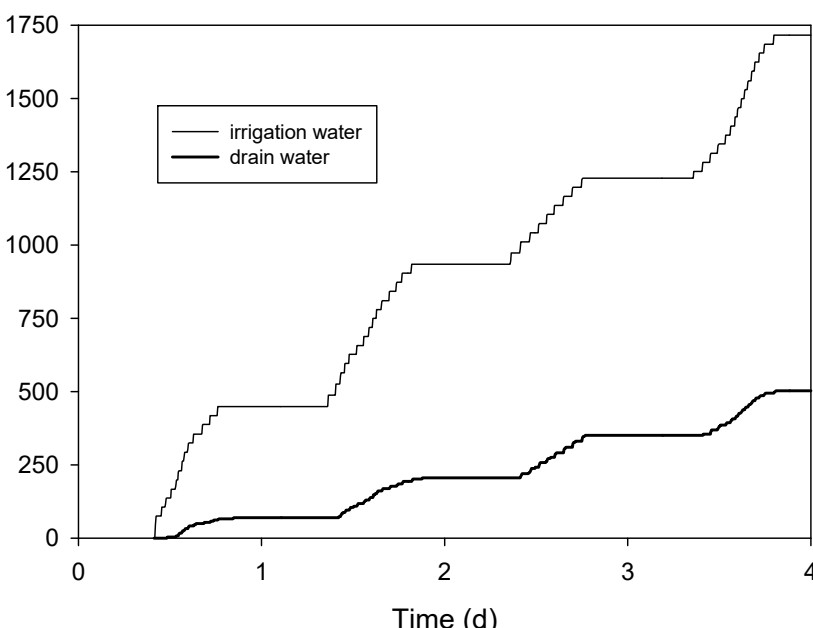

**Figure 4.** Cumulative irrigation and drain water volumes as a function of time as measured in the experiment. Irrigation is the flow from the mixing reservoir to the cultivation reservoir, and drainage is the flow from the cultivation reservoir to the filter reservoir. Time zero is 00.00 h 31 May.

Figure 5 shows that the time courses of the water content in the mats as measured with the two sensors were very similar. These time courses were strongly linked to the irrigation pattern: during the daytime, the water contents increased stepwise due to irrigation events followed by decreases until the next irrigation event; during the nighttime, there was a slow decrease. Using the average of the two water contents to estimate the time course of the water volume in the mats (as described before) resulted in a water volume in the cultivation reservoir ranging between 740 and 820 L. Figure 5 shows also that the two water contents

differed from each other by approximately 15%. In combination with the uncertainty in the water content of the blocks (see Model parameterization), we estimate that the uncertainty in the water volume of the cultivation reservoir (i.e., the 95% confidence interval) to be approximately ±25%. This uncertainty will be considered in the test of the model.

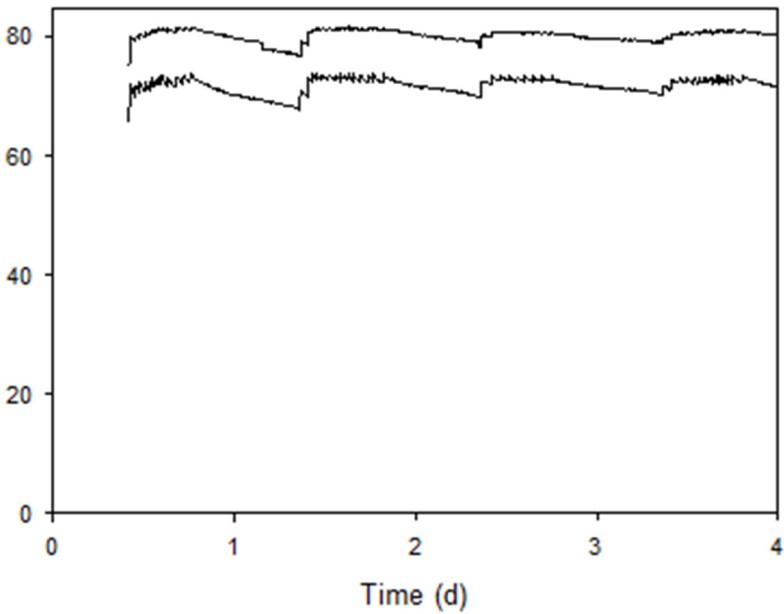

**Figure 5.** Water content (as % of pore volume) in the stone wool as a function of time as measured with duplicate sensors in two stone wool slabs. Time zero is 00.00 h 31 May.

On the basis of the added pesticide masses and the measured initial volume of the mixing tank, initial concentrations of imidacloprid and pymetrozine were expected to be 14.4 and 11.0 mg/L, respectively. Initial concentrations in the mixing tank were measured in duplicate within 6 min after application (before the first, manual started irrigation event) and were found to be 13.2 and 16.0 mg/L (average 14.6 mg/L) for imidacloprid and 8.6 and 10.7 mg/L (average 9.7 mg/L) for pymetrozine. So, for imidacloprid, the measured concentration was a few percentage points higher than expected, and for pymetrozine, it was approximately 10% lower. In view of the approximate 20% difference between the duplicate samples, it is likely that the mixing was not yet complete at the first sampling despite the thorough mixing of the water in the mixing tank. Later sampling times could not be used to check the dose because these took place after the first irrigation event.

Figure 6 shows that measured and simulated concentrations in the mixing tank corresponded quite well. Note that the horizontal axis does not denote time but rather the cumulative water volume that flowed out of the tank. This is chosen because this cumulative volume is the driving force for the decrease. This good correspondence was to be expected, as the uncertainty resulting from the model assumptions and the parameter values is quite small for this tank, i.e., the only relevant processes are perfect mixing and degradation. For imidacloprid, no degradation was assumed by using a half-life of 1000 d; for pymetrozine, a half-life of 8.5 d was assumed at 25 °C. In Figure 6, a cumulative volume of irrigation water of 1000 L corresponds to a time period of approximately 1 day (see also Figure 4), so degradation of pymetrozine hardly influenced these simulated concentrations. The possible incomplete mixing during the first sampling did not lead to an increased difference between measured and simulated concentrations.

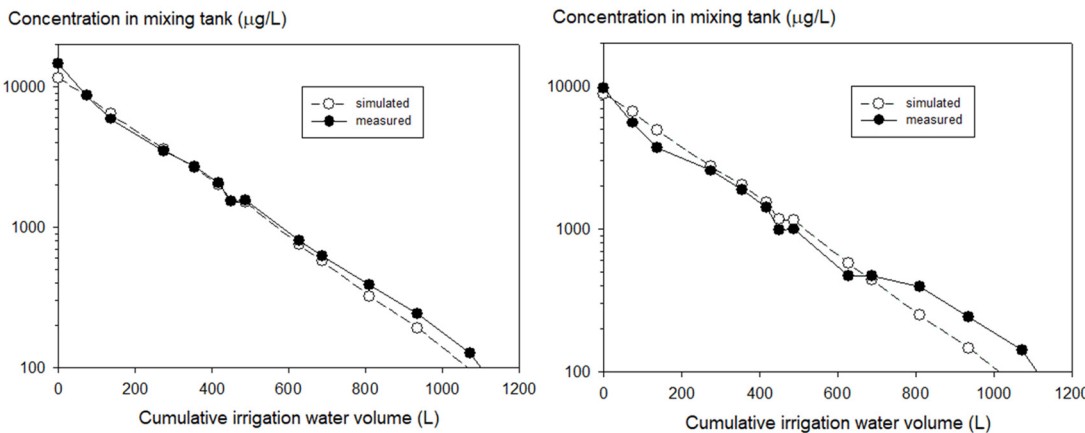

**Figure 6.** Measured and simulated concentrations of imidacloprid (**left**) and pymetrozine (**right**) in the mixing tank as a function of the cumulative volume of irrigation water, i.e., the water that was pumped out of the mixing tank. Irrigation occurred in batches every 2–3 h, and the mixing tank was refilled. Concentrations were measured in duplicate, and the average value is shown in the graphs.

Figure 7 shows that simulated breakthrough of both pesticides in the used-water reservoir was faster than measured and that simulated concentrations at the end of the model test were approximately two times higher than measured (we plotted here on the horizontal axis the cumulative volume of water that was discharged into this reservoir, as concentration changes in this reservoir are driven by this inflow). This factor of two is rather high for a model, especially when used in regulatory practice.

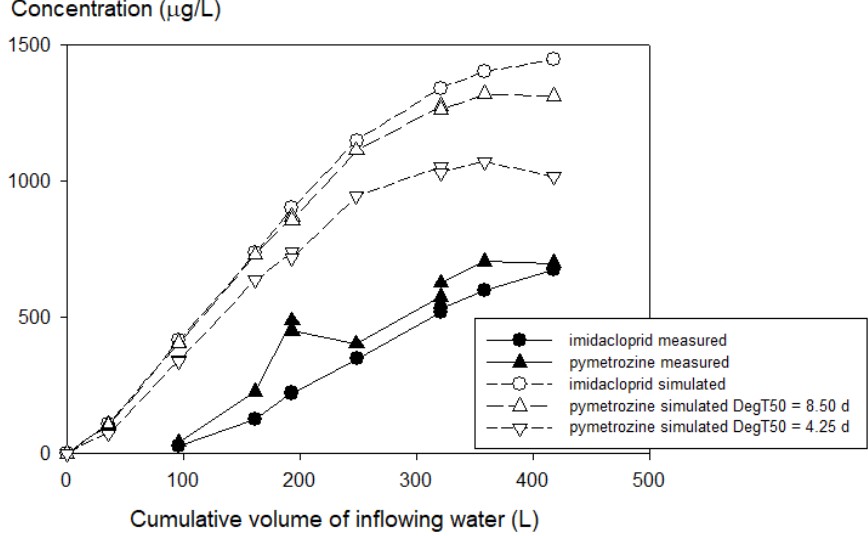

**Figure 7.** Measured and simulated concentrations of imidacloprid and pymetrozine in the used-water tank as a function of the cumulative water volume flowing into this tank. Simulations were based on first estimates for all parameters except the run with the shorter half-life for pymetrozine. Both measured and simulated concentrations of pymetrozine were multiplied by 2.94/2.25 to account for the difference in dosage between imidacloprid and pymetrozine.

Possible causes for the too-high simulated concentrations are: (i) more dilution in the cultivation reservoir than simulated due to an estimated too-low water volume in the mats, (ii) more plant uptake than simulated due to a too-low TSCF or partitioning into the plant roots (not included in the model, which considers only uptake due to transpiration), (iii) faster degradation in the rooted stone wool than simulated, (iv) significant sorption to the stone wool mats or transport pipes (sorption is not included in the model).The faster simulated breakthrough may also have been due to incomplete mixing in the cul-

tivation reservoir. Hereafter, we will consider these possibilities one by one and discuss their plausibility.

As described before, we consider the uncertainty in the water volume in the cultivation reservoir to be approximately 25%. Thus, we made indicative calculations assuming a 25% higher volume, as a higher volume will lead to lower calculated concentrations. Results in Figure 8 show that increasing the volume indeed led to a lower simulated concentration. However, the figure also shows that the uncertainty in the water volume of the cultivation reservoir is unlikely to be responsible for the poor performance of the model, as the effect is relatively small.

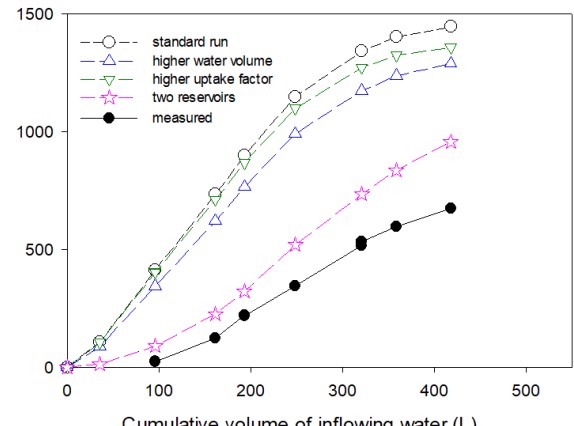
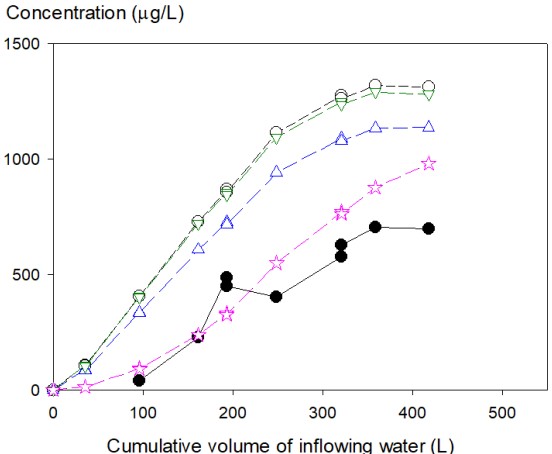

**Figure 8.** Measured and simulated concentrations of imidacloprid (**left**) and pymetrozine (**right**) in the used-water tank as a function of the cumulative water volume flowing into this tank. The standard run was based on first estimates of the parameters (also shown in Figure 7); the run with the higher water volume assumed a 25% increase in the volume of water in the cultivation reservoir; the run with higher uptake factor assumed a 25% increase in the TSCF; for the run with the two reservoirs, the cultivation reservoir was divided into two reservoirs of equal size with plant uptake only from the first reservoir. Both measured and simulated concentrations of pymetrozine were multiplied by 2.94/2.25 to account for the difference in dosage between imidacloprid and pymetrozine.

The possible effect of increased plant uptake was checked by performing calculations with a 25% higher TSCF (i.e., 0.54 instead of 0.43 for imidacloprid and 0.20 instead of 0.16 for pymetrozine). Figure 8 shows that this decreased the simulated concentrations only to a small extent. It was a priori already somewhat unlikely that the TSCF could be responsible for the discrepancies in view of the large difference between the two TSCFs (0.43 versus 0.16), while the discrepancies were similar for the two pesticides. Furthermore, increasing the TSCF did not lead to a slower breakthrough. Thus, it is unlikely that uncertainty in the TSCF was responsible for the difference between simulated and measured concentrations.

The model considers only plant uptake that is proportional to the transpiration rate using the TSCF concept. However, additionally plant uptake by partitioning into the roots will take place (this was not yet included in the model because only very limited information on the fresh root mass in stone wool mats was available). This partitioning can be described by the concept of the so-called root concentration factor (RCF [10]). This RCF is defined as the concentration in the roots (i.e., mass of pesticide in roots per mass of wet roots) divided by the concentration in the nutrient solution. Briggs et al. (1982) [12] established a relationship between the RCF and the $K_{ow}$, showing that the RCF increases with increasing $K_{ow}$. This relationship gives an RCF of 0.84 L/kg for pymetrozine and of 0.90 L/kg for imidacloprid. Assuming equilibrium between the roots and the solution,

Boesten and Matser (2017) [17] showed that the fraction of the total pesticide mass in the cultivation reservoir present in the roots ($f_r$) is given by

$$f_r = \frac{M\ RCF}{V + M\ RCF} \tag{3}$$

where $M$ is mass of wet roots (kg), and $V$ is volume of water (L) in the system. From measurements for a full-grown sweet pepper crop, they estimated that $M$ equals approximately 0.1 kg if $V$ is approximately 1 L. This gives an $f_r$ of 0.08 (i.e., 8%) for both pymetrozine and imidacloprid. Thus, it is unlikely that partitioning into the plant roots explains the difference between modelled and measured concentrations. For pesticides with a much larger $K_{ow}$, the partitioning into plant roots may have a considerable effect on simulated concentrations. However, such pesticides are unlikely to be applied with the irrigation water because their translocation to the above ground parts of the plants is very limited, as such pesticides have low TSCF values [12].

Boesten et al. (2018) [18] reviewed available information on degradation half-lives of pesticides in rooted stone wool growing systems and compared these with hydrolysis studies. They found reliable information for metalaxyl, oxamyl, dimethomorph, fluopyram, and imidacloprid. All these pesticides were stable in hydrolysis studies in the relevant pH range. For metalaxyl, half-lives of 5 and 6 d were found. For the other pesticides, only lower limits of the half-lives could be derived: much larger than 22 d for oxamyl and much larger than 6 d for the other three pesticides (including imidacloprid). So, it is unlikely that a faster degradation of imidacloprid (than assumed on the basis of hydrolysis) could explain the difference between simulated and measured drainage concentrations. However, in view of the short half-lives of metalaxyl, it is possible that the half-life of pymetrozine was considerably shorter than derived from the hydrolysis rates. So, we made a calculation assuming a half-life of 4.25 d (i.e., half the value used before and close to the half-live of metalaxyl). Figure 7 shows that this did not lead to a significant improvement of the description of the measurements for pymetrozine. So, both for imidacloprid and pymetrozine, it is unlikely that faster degradation in the rock wool can explain the differences between measured and simulated drainage concentrations.

Given that the efficacy of a pesticide depends on its availability, it is very unlikely that the overestimation of the concentration is due to sorption. To explore the potential impact of sorption on the calculated concentrations, we found that Boesten and Matser (2017) [15] measured the sorption of pymetrozine (with a $K_{ow}$ of 0.65) to Grotop stone wool and found a linear sorption coefficient of 0.2 L/kg. They estimated that this sorption would lead to a decrease in the concentration in the water in the stone wool mats of approximately 2%. Using sorption measurements from the literature of two pesticides ($K_{ow}$ 1000–10,000) and including their sorption measurement of dimethomorph ($K_{ow}$ 479) showed a positive correlation between the octanol water coefficient ($K_{ow}$) and sorption to stone wool. These three pesticides showed sorption coefficients between 1 and 2 L/kg. For dimethomorph, they estimated a decrease in the concentration in liquid phase due to sorption of 9%. The $K_{ow}$ of imidacloprid (3.7) is much closer to that of pymetrozine (0.65) than to those of these three pesticides (479–10,000). As a result, its sorption coefficient to stone wool is likely much closer to 0.2 L/kg than to 1 L/kg. Therefore, sorption of imidacloprid to the stone wool will likely lead to a concentration decrease that is only slightly higher than the 2% found for pymetrozine. Boesten and Matser (2017) [15] found that sorption of pymetrozine to the PVC transport pipes was unmeasurably small. It can be expected that this sorption is also related to the $K_{ow}$. As the $K_{ow}$ values of pymetrozine and imidacloprid are quite close, sorption of imidacloprid to the pipes is expected to be small as well. So, sorption to the stone wool or pipe materials is unlikely to be responsible for the poor performance of the model.

Assuming that the cultivation reservoir behaves as a perfectly mixed reservoir is the simplest approach possible, which was taken. Ideally the model would simulate the water flow and transport in each slab separately while considering the root distribution

in the slabs. The water and pesticide mass would then be collected in the troughs and the transport in the troughs simulated over time. Because, as yet, no studies are available on solute flow processes in stone wool mats grown with crops, we used a simplified model approach, assuming that the entire system of slabs, plants, tubes, and troughs could be simulated as a perfectly mixed reservoir. As a next step, we assumed that solute behaviour in the cultivation reservoir can be described with two sequential perfectly mixed reservoirs of equal size (50%–50%) with plant uptake from both reservoirs (i.e., the next most simple model. The final step in this series would then be to have an infinite number of interconnected reservoirs, each representing a part of the system). This decelerated the breakthrough and lowered the concentrations in the first 200 L of water flowing into the used-water tank but increased even the concentrations after approximately 250 L of water inflow, so this did not improve the correspondence between the measurements and the simulations. We then checked the influence of the size of the reservoirs, assuming that the first and second reservoirs had volumes of 83% and 17%, respectively, of the total cultivation reservoir with plant uptake rates proportional to the volume of the reservoir. This produced almost exactly the same result as the 50%–50% assumption. So, the seize of the two reservoirs had no significant effect on the breakthrough curve.

As a next step, it was tested whether the measurements can be described by inhomogeneous root uptake from the cultivation reservoir. Again, the cultivation reservoir was subdivided into two sequential perfectly mixed reservoirs of equal size, but now the plant uptake took place from only the first reservoir. The rationale behind this step is that the water with solutes first enters the part of the slabs with a higher abundance of roots, and in a next step, it leaches to the troughs and is transported to the next reservoir. So, in the second reservoir, there are no or limited roots to enable the plant uptake. Simulated concentrations in the first reservoir will be higher than those in the second reservoir. Improvement of the model description of the measured concentrations can be obtained only by higher plant uptake, so by assuming plant uptake from the first reservoir only. Simulations with plant uptake only from the second reservoir confirmed that this increased concentrations in the drainage water. Figure 8 shows that assuming plant uptake from only the first reservoir improved the correspondence between simulations and measurements considerably. However, the simulated concentrations are still approximately 25% too high. We checked whether a smaller size of the first reservoir could result in a better description. We did so for imidacloprid because its TSCF is much larger than that of pymetrozine (0.43 versus 0.16), so the effect of a change in plant uptake is expected to be higher for imidacloprid. Decreasing the volume of the first tank from 50% (i.e., equal size) to only 20% of the total cultivation reservoir volume decreased the concentration after approximately 400 L of water inflow from approximately 960 µg/L (Figure 8) to approximately 880 µg/L, so it was still much larger than the measured concentration of approximately 700 µg/L (see Figure 8).

## 4. Discussion and Conclusions

Testing of the SEM model against presented experimental data showed that although the concentration in the mixing reservoir were relatively well predicted, the concentrations in the waste-water reservoir were poorly predicted. The simulation of the concentrations draining from the stone wool mats could be considerably improved by assuming that the behaviour in the mats can be represented by two sequential perfectly mixed reservoirs with plant uptake only from the first reservoir. One mechanism behind this assumption may be that plant roots are more abundant in the regions where the irrigation water enters the mats because nutrient concentrations will be highest in these regions. Another mechanism may be that there are preferential flow paths of the water in the mats and that roots are concentrated in these flow paths because these paths contain the highest nutrient concentrations. The irrigation water dripped into the three planting blocks (7.5 cm high) whose centres were 40 cm apart, whereas the height of the mats below is only 7.5 cm. It seems likely that, at this point, irrigation leads to downward water flow rates in the mats

(driven by gravity) that are faster in the region below the drippers than in the region in the middle between the drippers. So, this may be a driver for the occurrence of preferential flow.

The irrigation volume was typically 30 L per unit, whereas the total water volume in the mats was 720–840 L. So, each individual irrigation resulted in increases in the water content of the mats of only a few percentage points (as also illustrated by Figure 5), whereas these small increases, nevertheless, resulted in drainage of approximately 30% of the irrigation volume. It seems probable that such small increases led to a solute flow pattern that can be described better by assuming a convection–dispersion model (i.e., a water-displacement model) than by assuming perfect mixing. The numerical solution of a convection–dispersion model is commonly obtained by a series of numerical layers that are each perfectly mixed. So, assuming two sequential reservoirs instead of one is a step in the direction of a convection–dispersion model.

Incomplete mixing of the water in the mats in combination with inhomogeneous water uptake by roots was likely the main cause of the differences between measured and simulated concentrations. However, this conclusion is based on indirect evidence: after elimination of other likely causes of the discrepancies, this combination led to a significant improvement of the description of the measurements. The new version of GEM now represents the cultivation part of the greenhouse by two sequential perfectly mixed reservoirs. It may be tempting to develop a more sophisticated solute flow model for such rooted stone wool mats. However, given the limited data available, we recommend doing so hand in hand with experiments aimed at giving direct evidence and data that quantify and underpin the parameters used in the solute flow model. These experiments could include, for example, slicing the mats into layers and measuring the inhomogeneity of the pesticide concentrations and the plant roots and visualising the water flow paths by adding a coloured tracer to the irrigation water. In addition, measuring the concentrations in the individual drippers and at various location in the troughs would then be advised.

As described before, adding equilibrium partitioning into plant roots using the concept of the root concentration factor RCF (based on a single measurement of the fresh root mass) would have decreased simulated concentrations in the homogeneous cultivation reservoir by approximately 8%. Splitting this reservoir into two equal parts with plant uptake restricted to the first part will likely increase the effect of root partitioning on simulated concentrations to levels above 10% for these two pesticides. It seems advisable, therefore, to include partitioning into plant roots in the model. This will require collection of data on fresh root masses in stone wool mats, as these are hardly available. It is recommended to include this process in addition to the above-described options to account for the inhomogeneous water uptake by the plants.

Since the GEM model is used in the regulatory risk assessment for pesticides, confidence in the model is a prerequisite. This test made clear that testing GEM against experimental data is needed to increase the confidence in the model and to assess and understand which processes are driving the concentration of the recirculation water with the final aim of improving the model concepts. As a first step, we recommend the more detailed experiment as suggested above. In a next step, other application methods, e.g., spraying or low volume misting (LVM), should be considered. For these application methods, additional processes play a role; for example, they determine the entry of the substances in recirculation water, such as deposition on various surfaces in the greenhouse and volatilization. For extending the GEM model application to greenhouse systems in other countries, e.g., Spain or Sweden, it will be worthwhile to assess variants with an open system without recirculation, as emission to surface water will be much higher in such systems.

**Author Contributions:** Conceptualization, E.L.W.; Methodology, E.L.W., J.J.T.I.B., E.A.v.O. and W.H.J.B.; Software, W.H.J.B.; Data curation, E.A.v.O.; Writing—original draft, E.L.W. and J.J.T.I.B.; Writing—review & editing, E.L.W., E.A.v.O. and W.H.J.B.; Visualization, J.J.T.I.B. and E.A.v.O. All authors have read and agreed to the published version of the manuscript.

**Funding:** This research was conducted as part of project BO-43-011-01-005, which is a policy-supporting project funded by the Dutch Ministry of Agriculture, Nature and Food Quality.

**Data Availability Statement:** The experimental dataset used in this manuscript is available via [11].

**Acknowledgments:** The authors thank Ton van der Linden and Martine Hoogsteen from the National Institute for Public Health and the Environment, the Netherlands and Laura Buijse, and Marieke van der Staaij from Wageningen Research for their contribution to the experiment and the analysis.

**Conflicts of Interest:** The authors declare no conflict of interest.

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
