# Peer review of "Testing the Greenhouse Emission Model (GEM) for Pesticides Applied via Drip Irrigation to Stone Wool Mats Growing Sweet Pepper in a Recirculation System"

_horticulturae, doi:10.3390/horticulturae9040495_

Round 1

Reviewer 1 Report

Very interesting study and potentially a very useful model for growers. Well-executed study. Would be great to converse more with the authors on this study and topic. I have a few comments and recommended edits in the attached document.

Author Response

dear reviewer, thank you for your valuable comments and suggestions for improvement, we incorporated many of them and gave arguments when we decided to take a different approach. We added our reactions to your comments in the reviewed document  for convenience.

the summarizing comments I would like to respond to below 

kind regards, Louise Wipfler

(i) If I understand correctly, the GEM predicted a much earlier breakthrough and more breakthrough of chemicals than measured. Only ~50% of the chemicals applied seemed to be recovered in the used water reservoir. The authors main hypothesis is that ~50% of the applied chemicals remained sequestered in the substrate, where they either stayed, were eventually transformed or degraded, or taken up by the plants. If this is accurate, for me this discrepancy is the main result in addition to GEM accuracy. I would like to see the authors more clearly state this in the abstract and conclusions (if true).

This is not entirely correct. we found a discrepancy between the model predictions and the measurements. we could exclude though that sorption or degradation was the main cause of the difference. We found as the best explanation that the representation of the cultivation part of the greenhouse as an ideally mixed reservoir needed reconsideration. Therefore we decided to represent it by two mixed reservoirs. We provide explanation and rationale in the paper for this approach.

(ii)Also a main conclusion is that half of the chemical applied was leached and degraded by the ozone system. This seems inefficient, but perhaps not? There could be more discussion regarding the amount of pesticides that leach versus remain in the substrate-plant system.

The chemicals were fully removed by the ozone installation and not by leaching. The ozone installation is used for removal of pathogens but has as a side effect the removal of the substances. This only occurs after the treatment, so it has no effect on the efficacy of the treatment

(iii)Thorough evaluation and simulation was conducted to explain the poor model prediction. However, a clear statement regarding the likely fate of the unaccounted 50% is lacking. The authors suggest non-uniform mixing in the cultivation reservoir, but also suggest adsorption potential is very low. Did the chemicals remain in the substrate solution to be later absorbed or degraded? A clear concluding statement in the abstract and results/discussion is needed on the likely fate of the chemicals and/or next steps to determine the fate for improving the model.

(iii) we changed the last two sections of the paper. We think that the message that we want to transfer is now better explained

Reviewer 2 Report

In this manuscript entitled “Testing of the Greenhouse Emission Model (GEM) for pesticides applied via drip irrigation to stone-wool mats growing sweet pepper in a recirculation system”, GEM was tested using an experiment in which imidacloprid and pymetrozine were applied via drip irrigation to stone-wool mats growing sweet pepper. The research is in-depth and detailed, which is of great significance to surface water quality. In general, this manuscript can be accepted for publication in this journal after minor revisons.

1. GEM was tested using an experiment in which imidacloprid and pymetrozine were applied via drip irrigation to stone-wool mats growing sweet pepper in this manuscript. The stability of imidacloprid and pymetrozine in the environment is also very important, and it is recommended that relevant literature should be added in the section of INTRODUCTION.

2. For the convenience of readers, it is recommended to add the structures of imidacloprid and pymetrozine.

3. Have the authors considered the impact of different growth cycles on the results of plants?

4. Figure 6 is ambiguous and needs adjustment.

5. The analyses were performed on an Agilent 1260 Infinity liquid chromatograph coupled with an 6460 Triple quad mass spectrometer (LC-MSMS).  The analysis of imidacloprid and pymetrozine requires more detailed data, such as mobile phase flow rate and detection wavelength, etc. 

Author Response

dear reviewer 2, thank you for your feedback and for sharing your suggestions for improvement. We responded to your comments in the uploaded file,

kind regards, Louise Wipfler
